# Structural basis of *Acinetobacter* type IV pili targeting by an RNA virus

Ran Meng [1,3,6], Zhongliang Xing [1,6], Jeng-Yih Chang[1,4,6], Zihao Yu[1], Jirapat Thongchol[1], Wen Xiao[1], Yuhang Wang [1], Karthik Chamakura [1,5], Zhiqi Zeng[1], Fengbin Wang [2], Ry Young [1], Lanying Zeng [1] & Junjie Zhang [1] ✉

Acinetobacters pose a significant threat to human health, especially those with weakened immune systems. Type IV pili of acinetobacters play crucial roles in virulence and antibiotic resistance. Single-stranded RNA bacteriophages target the bacterial retractile pili, including type IV. Our study delves into the interaction between *Acinetobacter* phage AP205 and type IV pili. Using cryo-electron microscopy, we solve structures of the AP205 virion with an asymmetric dimer of maturation proteins, the native *Acinetobacter* type IV pili bearing a distinct post-translational pilin cleavage, and the pili-bound AP205 showing its maturation proteins adapted to pilin modifications, allowing each phage to bind to one or two pili. Leveraging these results, we develop a 20-kilodalton AP205-derived protein scaffold targeting type IV pili in situ, with potential for research and diagnostics.

*Acinetobacter* infections impose a significant clinical burden, affecting various body systems, including lungs, bloodstream, and urinary tract[1]. The severity of these infections ranges from mild to acute, with immunocompromised individuals being most susceptible. The emergence of antibiotic-resistant strains, especially of the species *A. baumannii*, has significantly limited treatment options, necessitating the exploration of alternative therapeutic approaches, such as phage therapy[2].

Acinetobacters have type IV pili (T4P), which are involved in twitching motility, surface adhesion, biofilm formation, and natural transformation in gram-negative bacteria[3]. These thin, filamentous, and retractile appendages underlie the virulence and ability by which acinetobacters acquire and incorporate genetic material from the environment. Consequently, this results in the facile spread of antibiotic resistance genes, substantially increasing the challenges for bacterial treatment[4]. T4P is assembled by the type IV pilus system (T4PS), which is homologous to the type II secretion system (T2SS). While T4PS is responsible for assembling T4P, T2SS assembles a pseudopilus assumed to function as a 'piston' to facilitate protein secretion[5].

Single-stranded RNA bacteriophages (ssRNA phages) are positive-sense RNA viruses that have shown specificity for gram-negative bacteria by exploiting host retractile pili, including T4P[6]. They are lytic phages and their genomes, typically 3-4 kb long, exhibit a conserved organization of three core genes. These are, from the 5′ end, *mat*, *coat*, and *rep*, encoding the maturation protein (Mat), the coat protein (Coat) and the β-subunit of an RNA-dependent RNA replicase (Rep), respectively. In addition, there is usually a gene for a lysis protein, but its location is variable and often embedded out of frame for one of the core genes[7]. Recently, metatranscriptomic studies have discovered 65,814 sequences of these ssRNA viruses, with more than 12,000 sequences being complete or near-complete genomes, each containing three core genes[8]. While the structures of two model ssRNA phages, MS2[9–11] and Qβ[12], which infect *E. coli* via the conjugation F pilus[13], have been fully characterized, it is unclear if all ssRNA phages share the same structural features. Recently, studies have demonstrated that MS2 has the ability to bind to and induce detachment of the conjugative F-pili from infected *E. coli* cells[14]. As retractile pili play crucial roles in the virulence of many bacteria, in principle, if the

---

[1]Center for Phage Technology, Department of Biochemistry and Biophysics, Texas A&M University, College Station, TX 77843, USA. [2]Department of Biochemistry and Molecular Genetics, Heersink School of Medicine, University of Alabama at Birmingham, Birmingham, AL 35294, USA. [3]Present address: Yale University, New Haven, CT 06520, USA. [4]Present address: UMass Chan Medical School, Worcester, MA 01655, USA. [5]Present address: Armata Pharmaceuticals, Inc., Marina del Rey, CA 90292, USA. [6]These authors contributed equally: Ran Meng, Zhongliang Xing, Jeng-Yih Chang. ✉e-mail: junjiez@tamu.edu

infection-dependent pili detachment is a general feature for ssRNA phages, these simple viruses could be used to suppress the deployment of retractile pili and thus reduce the virulence of many important bacterial infections. Previously, we have determined the cryo-electron microscopy (cryo-EM) structure which revealed how MS2 binds to the F-pilus[15]. However, due to the high diversity of retractile pili in gram-negative bacteria, the structural mechanism obtained from one phage-pilus pair cannot apply to another.

In this study, we aim to investigate how the ssRNA phage, AP205, infects the host *acinetobacter* via T4P. However, there is limited structural information. Currently, only the structure of AP205 Coat has been solved, revealing a circular permutation compared to the Coat of MS2 and Qβ[16], whereby the proteins have a changed order of amino acids in their peptide sequences, while the overall three-dimensional shapes are similar. As for the receptor pili, although the recombinant globular domain of *A. baumannii* type IV pilin has been determined using X-ray crystallography[17], it may not accurately represent potential post-translational modifications of the native *Acinetobacter* T4P. Additionally, the molecular mechanism by which AP205 interacts with T4P is not yet understood. In this study, we determined cryo-EM structures of AP205 alone, *Acinetobacter* T4P alone, and AP205 in complex with T4P, which guided our design of a small protein scaffold to effectively target and label T4P in live cells.

## Results

### The complete structure of the AP205 virion

About 8% of the vitrified AP205 particles show a well-defined three-dimensional (3D) conformation of the genomic RNA (gRNA) inside the capsid. The smaller proportion of particles exhibiting a defined RNA organization, in contrast to MS2 (>90%)[18] and Qβ (~30%)[19], may be due to the more heterogeneous RNA conformations inside the capsid.

Nonetheless, we achieved a 3.1Å-resolution asymmetric reconstruction of the mature AP205 virion (Supplementary Fig. 1), enabling us to construct a complete atomic model of both the capsid protein shell and the gRNA, which consists of 4,269 nucleotides (Fig. 1a and Supplementary Movie 1).

In our structure, 178 copies of the Coat self-assembled into a near-icosahedral shell (with the triangulation number, *T*=3), measuring approximately 290 Å in diameter. Surprisingly in contrast to MS2 and Qβ, which have a single Mat per virion, AP205 possesses a distinctive Mat-dimer, resembling the shape of the letter "F" at a pseudo two-fold axis of the capsid and increasing the length of the virion for an additional 70 Å. The 3' region of the gRNA is proximal to the Mat-dimer on one side of the capsid, while the 5' region resides on the opposite side. This is consistent with the previously reported structures of ssRNA coliphages[9,20]. Notably, one RNA stem-loop originating from the 3' end of the AP205 gRNA, which interacts with the Mat-dimer, is partially exposed outside the capsid shell (Fig. 1a bottom). This observation strongly suggests that this specific RNA fragment is the initial region to exit the capsid and enter the cell.

Mat of AP205 is significantly larger (Supplementary Fig. 2a), consisting of 534 amino acids (aa) (393 aa in MS2 and 421 aa in Qβ)[9,15,21]. It can be divided into three domains, namely the apical, central, and basal domains (Fig. 1b, c). The apical domain consists of two small β-sheets, S1 (involving strands β2, β3, β9, β10, and β11) and S2 (involving strands β1, β4, β7, β8, and β12) connected by an α-helix (α1), while the central domain features a large β-sheet S3 (involving strands β5, β6, β13, β14, β15, β16, β17, and β18) and two short α-helices (α2 and α3). The basal domain comprises four α-helices (α4, α5, α6, and α7). The basal domains of the Mat-dimer insert into the capsid, while the apical domains are exposed on the surface. The two copies of Mat exhibit different conformations: One Mat (referred as outer Mat) extends

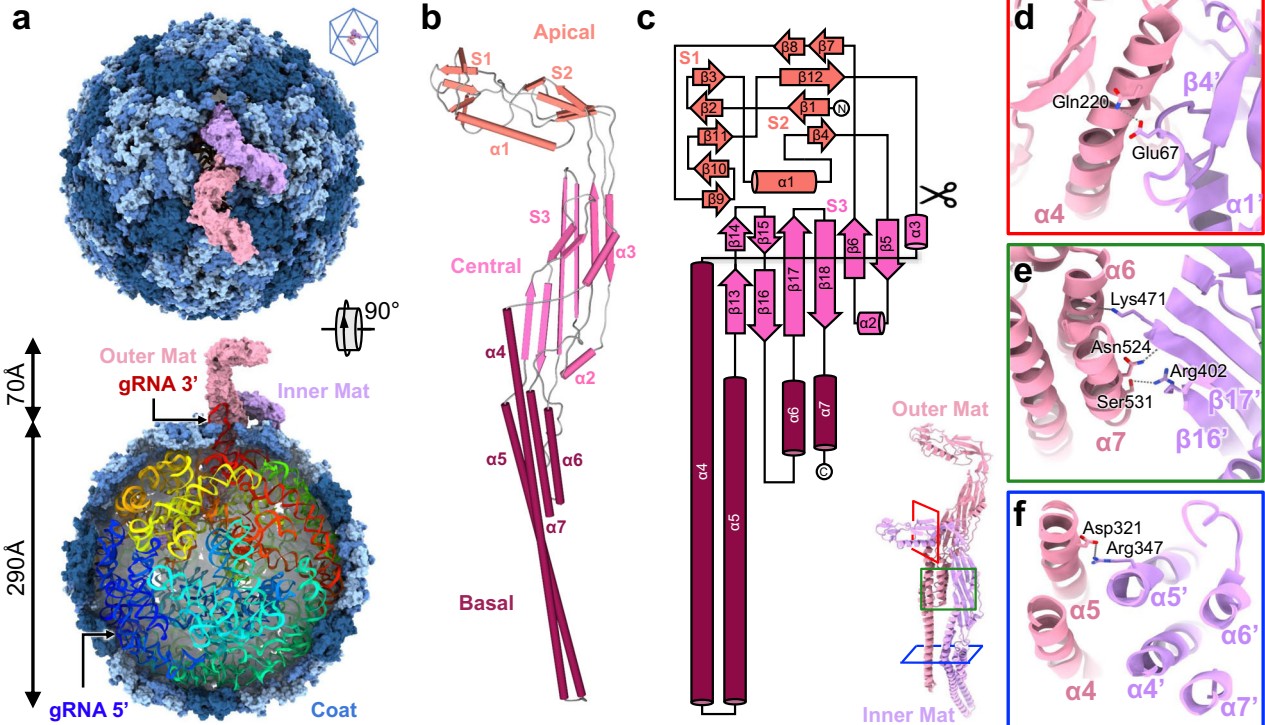

**Fig. 1 | The complete structure of AP205 mature virion encompassing a Mat-dimer. a** Top and cut-in views of AP205, showing Coat in different shades of blue, inner Mat in purple, and outer Mat in pink. The gRNA is rainbow-colored. The cage lattice (top right) indicates the location of the Mat-dimer in the capsid, which breaks the icosahedral symmetry at a two-fold axis of the Coat shell. **b** The model of the Mat displaying three domains: apical, central, and basal. **c** The cartoon representation for the secondary structure of the Mat. The scissor sign indicates the truncation to generate the protein scaffold, Mat200. **d–f** Three subunit interfaces within the Mat-dimer. The red, green, and blue boxes on the dimer model (lower left) indicate the zoomed-in regions for Panels **d** (viewing from left), **e** (viewing from front), and **f** (viewing from top), respectively.

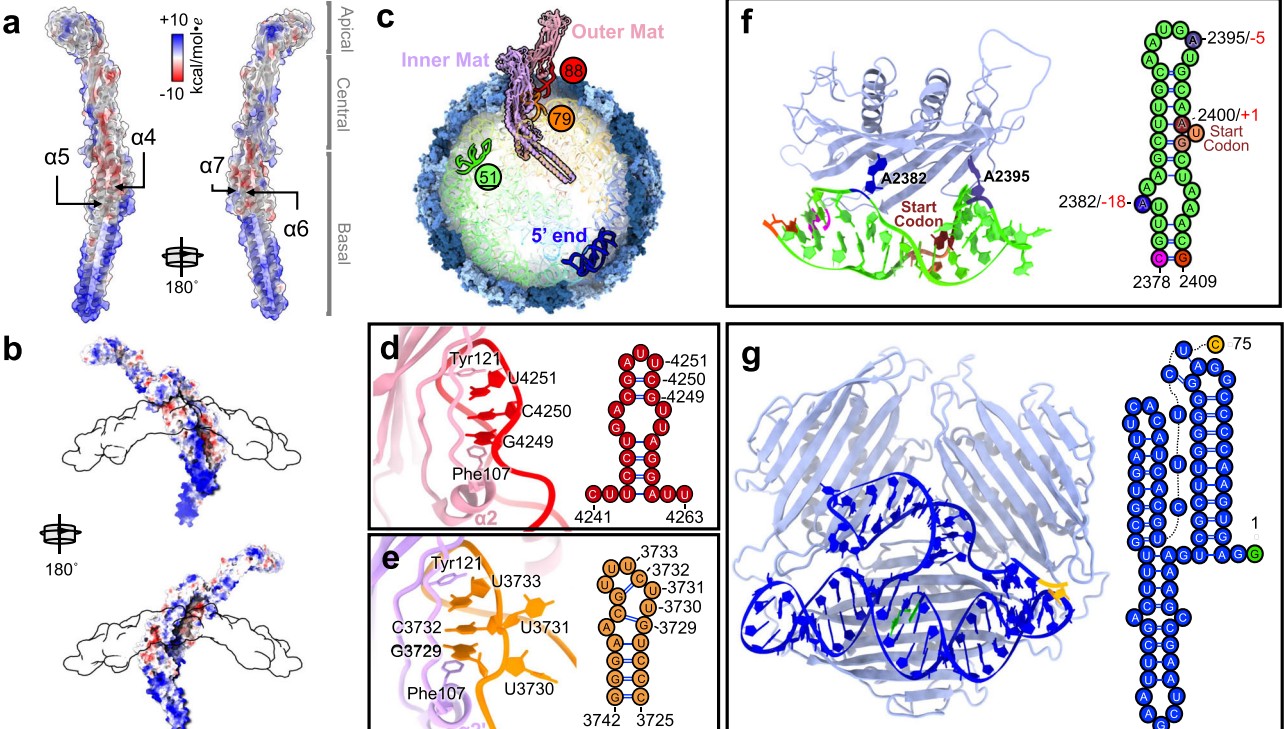

**Fig. 2 | Interactions between capsid proteins and the gRNA. a** Electrostatic potential map of a single Mat (red: negatively charged; blue: positively charged). **b** Electrostatic potential map of the Mat-dimer inserted in the Coat shell (white transparent surface). **c** Locations of Stem-loops 79, 88, and 51 (the operator, underlined), and the first 75 nucleotides of the 5′ end in the virion. **d**, **e** Models and secondary structures of Stem-loops 88 (Panel **d**) and 79 (Panel **e**), which interact with outer and inner Mat, respectively. **f** The model and secondary structure of Stem-loop 51, which interacts with one Coat-dimer. The start codon AUG, −5, and −18 positions are colored and labeled. **g** The model and secondary structure of the first 75 nucleotides at the 5′ end of the gRNA, which interacts with three Coat-dimers.

further to elevate its apical domain away from the capsid shell while the other Mat (referred as inner Mat) is embedded deeper within the capsid and has a larger bending angle between the central and basal domains (Supplementary Fig. 2b and Supplementary Movie 2). These two Mat proteins intertwine to form hydrogen bonds and salt bridges at three sites. The first interaction occurs between the α4 helix from the outer Mat and the loop connecting β4 and α1 from the inner Mat, involving a hydrogen bond between Gln220 (outer Mat) and Glu67 (inner Mat) (Fig. 1d). The second interaction involves α6/α7 from the outer Mat and β16/β17 from the inner Mat. It includes a hydrogen bond between Ser531 (outer Mat) and Arg402 (inner Mat), as well as two hydrogen bonds of Asn524 (outer Mat) and Lys471 (inner Mat) with the protein backbones of the opposite Mat (Fig. 1e). The third interaction is within a six-helix bundle between the two basal domains, connected by a salt bridge between Asp321 (α5 of the outer Mat) and Arg347 (α5 of the inner Mat) (Fig. 1f). These interactions hold the two copies of AP205 Mat together, forming a dimer that nearly triples the volume and doubles the surface area compared to a single Mat in ssRNA coliphages.

### Interactions between the gRNA and capsid proteins

Each Mat monomer presents large negatively charged protein surface patches around the top half of the basal domain (Fig. 2a and Supplementary Movie 3–5). These patches are formed by acidic amino acid sidechains present on Helices α6, α7, and the upper portion of Helices α4 and α5, which connect to the central domain. However, due to the negatively charged nature of the phage gRNA phosphate backbone, such an acidic surface is not favored to interact with the RNA, particularly for the inner Mat that penetrates deeper into the capsid. The dimerization of the two Mat proteins resolves this issue by shielding the negatively charged surface of the inner Mat with the lower part of

the basal domain from the outer Mat. This results in a positively charged Mat-dimer surface in the capsid-embedded regions of the basal domains (Fig. 2b). Concurrently, the corresponding negatively charged region in the outer Mat is protected by the coat protein shell (Fig. 2b and Supplementary Fig. 3). This arrangement creates a more energetically favorable Mat-dimer configuration that is compatible with virion assembly.

The AP205 gRNA is highly branched, presenting a total of 88 stem-loops (Supplementary Fig. 4), two of which provide specific interactions with the Mat-dimer (Fig. 2c) via multiple π-stackings between aromatic amino acid sidechains and RNA bases. The first site for the π-stacking takes place between the central domain of the outer Mat and the last RNA helix, Stem-loop 88 of the gRNA. Here, the sidechains of Phe107 from α2 and Tyr121 from the following loop stack with the bases G4249, C4250, and U4251 (Fig. 2d). Another π-stacking site occurs in the inner Mat at the corresponding location, between the central domain and Stem-loop 79. In this site, the corresponding Phe107 and Tyr121 of the inner Mat stack with G3729, C3732, and U3733 (Fig. 2e). A comparison between Stem-loops 88 and 79 reveals a remarkable similarity in RNA motifs, specifically at the tip of the RNA stems. Stem-loop 88 has the sequence "4249-**G**C**U**UAGC-4255", while Stem-loop 79 features "3729-**G**UU**C**UUUGC-3737". Notably, the two extra uridines (underscored) in the above sequence of Stem-loop 79 are flipped out, enabling two base pairings between C and G, similar to the corresponding site in Stem-loop 88 (Fig. 2d right and e right). The G and U (bold font) in the above sequence are the two bases where aromatic sidechains Phe107 and Tyr121 clamp onto. This strongly suggests the presence of a conserved "GCUUXGC" motif that has undergone evolutionary optimization to facilitate a strong interaction with each of the two Mats.

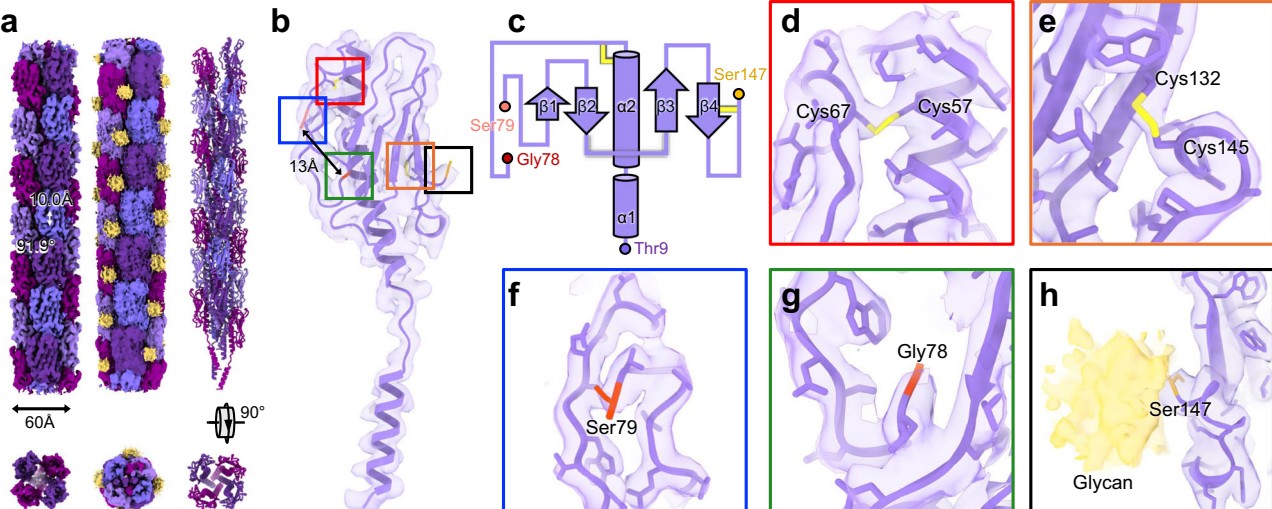

**Fig. 3 | The structure of *Acinetobacter* T4P. a** The cryo-EM density map of *Acinetobacter* T4P displayed at an isosurface threshold of 0.065σ with diameter, helical rise and helical twist labeled (left column). The same density map displayed at a lower isosurface threshold of 0.025σ, which shows the density for glycans (yellow) on the pilus (middle column). The atomic model of *Acinetobacter* T4P (right column). **b** The atomic model of one pilin. Colored boxes indicate the zoomed-in regions in Panels (**d–h**). The distance between Gly78 and Ser79 is around 13 Å. **c** The cartoon illustration for the secondary structure of the *Acinetobacter* Type IV pilin. **d** and **e** The model and map showing disulfide bonds of Cys57/Cys67 (Panel **d**) and Cys132/Cys145 (Panel **e**), respectively. **f** The cryo-EM density around Ser79. **g** The cryo-EM density around Gly78. **h** The zoomed-in view of the glycosylation site at Ser147. The glycan density (yellow) is weaker than the protein density and displayed at an isosurface threshold of 0.025σ.

Previous investigations have demonstrated that in ssRNA phages, the gRNA engages with the inner surface of the Coat shell through stem-loop structures. These Coat-RNA interactions are exemplified by the binding of a Coat-dimer to the RNA operator for the *rep* gene[22], where the base of adenosines, around the tip of the stem-loop, flips out to interact with the Coat-dimer. During later stages of ssRNA phage infection, Coat-dimers bind to the RNA operator to suppress the expression of the replicase protein, controlling the production of their progeny within the host cell, particularly when an excessive number of Coat proteins have been synthesized[23]. The RNA operator stem-loop for AP205 (Stem-loop 51) stands out as notably longer compared to other known ssRNA phages (Supplementary Fig. 5), resulting in a distinct interaction with the Coat-dimer. Our structural analysis revealed two adenosines at positions −5 and −18, relative to the start codon of the *rep* gene, flip out to interact with a Coat-dimer, sequestering the start codon of *rep* from the translation machinery (Fig. 2f).

Among the 88 stem-loop structures found in the AP205 gRNA, 13 of them were clearly discernible in our cryo-EM map, indicating their strong affinity for the capsid shell (Supplementary Fig. 6a–l). Previous research has proposed that these stem-loops, many of which exhibit flipped-out adenosine residues, potentially function as "packaging signals" that aid in the folding and encapsulation of the gRNA[24]. Notably, in AP205, numerous interactions between gRNA and Coat deviate from the typical packaging signal forms. In some cases, there are no flipped adenosines to interact with the Coat. Particularly, the 5' end of the gRNA (Residues 1-75) folds into a distinct three-way junction domain, wherein the last few nucleotides in this domain form a pseudoknot with the first RNA stem-loop (Fig. 2g). This RNA domain engages in interactions with three adjacent Coat-dimers and exhibits well-defined density, indicating a stable association with these Coat-dimers. However, no flipped adenosines are observed in this RNA structure. It is the unique shape of this RNA domain that allows it to fit precisely into the positively charged interface formed by the three Coat-dimers (Supplementary Fig. 6m, n). Given its location at the beginning of a newly synthesized gRNA, this RNA domain may recruit the initial Coat-dimers to seed the assembly of the capsid shell. Notably, the 5' half of the gRNA has very few RNA stems that specifically

interact with the capsid (Supplementary Fig. 7). This may contribute to more heterogeneous gRNA conformations when being packaged, leading to a much lower percentage of the AP205 particles showing defined gRNA density. It is worth noting that approximately 1% of the particles from our dataset produce a reconstruction that have a larger diameter of 310 Å and exhibit a *T=4* icosahedral symmetry (Supplementary Fig. 1 and Supplementary Fig. 8), increasing the capsid volume by ~20%.

## Structure of the *Acinetobacter* T4P

The host for AP205 is the *Acinetobacter genomosp. 16* (A. gp16, NCBI:txid70347), which is homologous to *A. baumannii*, with a genome sequence similarity of 87% over 73% coverage of the *A. baumannii* TP1 chromosome. Electron microscopy shows that these *Acinetobacter* cells have two types of morphologies for the cell surface appendages: straight and curved (Supplementary Fig. 9a). Density gradient-based purification is ineffective at separating these appendages, but mass-spectrometry analysis indicates that they are T4P and a pilus of unknown functions (annotated as FilA), respectively. Cryo-EM images reveal that AP205 particles bind to the straight T4P, but not to the curved FilA filament (Supplementary Fig. 9b). Using single-particle cryo-EM and helical reconstruction, we obtain a high-resolution structure of the purified straight T4P at 2.5 Å resolution (Fig. 3a and Supplementary Fig. 10) and model the mature pilin into the cryo-EM density (Fig. 3b). The sequence of the signal peptide, consisting of the first seven amino acids, is exclusively found in the prepilin and absent in the mature pilin[25].

The mature T4P pilin subunits have a distinct "lollipop" shape, presenting a globular C-terminal β-sheet domain and two N-terminal α-helices (Fig. 3. b and c). They assemble into a 60Å-wide filament, with a helical rise of 10.0 Å and a twist angle of 91.9°. The β-sheet domain is exposed on the outside, while the α-helices are buried within the center of the filament.

In the related species *A. baumannii*, T4P pilins undergo glycosylation mediated by a tfpO-like O-oligosaccharyltransferase[26]. Within our helical reconstruction of *A. gp16* T4P, additional density corresponding to potential glycans was observed (Fig. 3a, h).

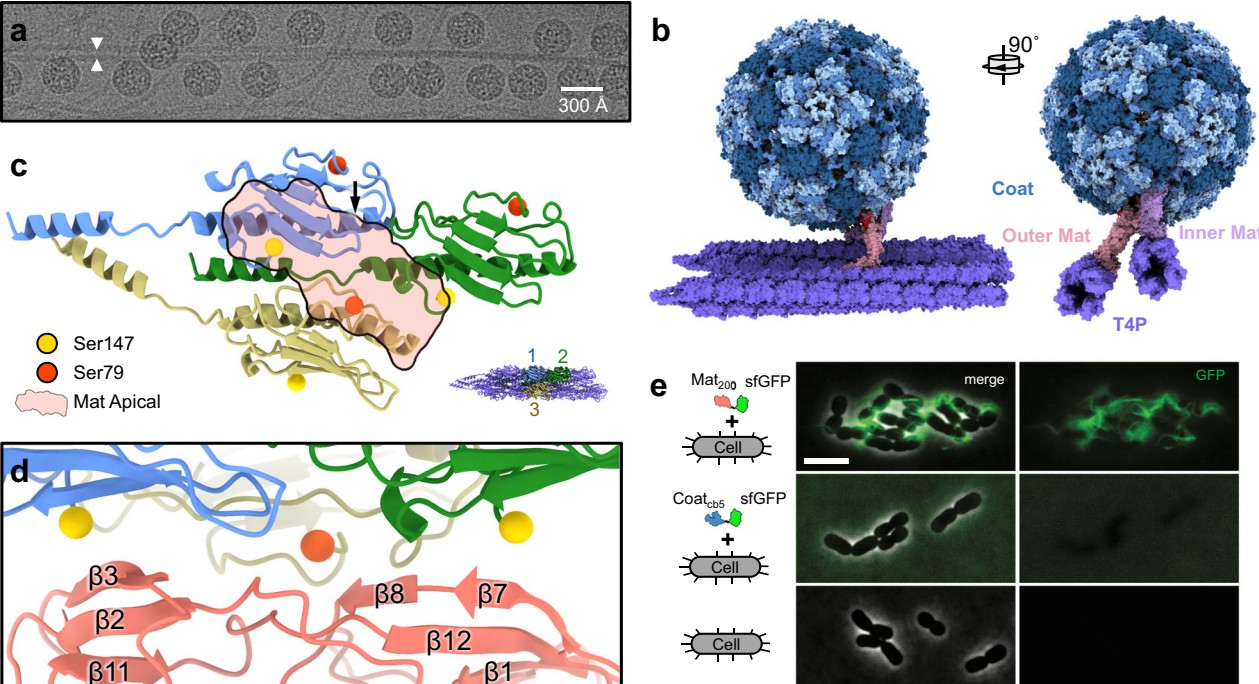

**Fig. 4 | Binding of AP205 to *Acinetobacter* T4P and the design of Mat200. a** One representative region of a raw micrograph out of 12,703 micrographs which all show AP205 phage particles adsorbed to *Acinetobacter* T4P. The white arrowheads indicate two adjacent T4P. The scale bar denotes 300 Å. **b** The cryo-EM structure of AP205 bound to two T4P. **c** The apical domain of each Mat directly interacts with three adjacent pilins (Pilin 1: blue; Pilin 2: green; Pilin 3: yellow). The red transparent region indicates the "footprint" of the apical domain on T4P. The orange and yellow spheres represent Ser79 and the C-terminal Ser147, respectively. The model in the

bottom right corner indicates the locations of these three pilins involved in the interaction with a Mat. **d** The zoomed-in view of the interface between AP205 Mat and *Acinetobacter* T4P. **e** Fluorescence light microscopy showing *A. gp16* T4P specifically labeled by Mat200-sfGFP. Top row: The micrograph of Mat200-sfGFP binding to T4P of *A. gp16* cells. Middle row: The micrograph showing Coat_cb5-sfGFP does not bind to *A. gp16* cells. Bottom row: The micrograph of *A. gp16* cells alone. At least 3 replicates are performed independently and they show similar results. The scale bar denotes 5 μm.

These glycans are added post-translationally to the last amino acid Ser147 of the mature pilin. Previous studies have indicated that glycosylation of pili assists *A. baumannii* in biofilm formation and virulence[27]. AP205 retains the ability to bind to *A.gp16* T4P (Supplementary Fig. 9b, c). The cryo-EM density of the glycan is weaker than the protein density. This might be attributed to the flexible attachment of the glycan to Ser147, glycan constitutional heterogeneity, and substoichiometric glycosylation of pilins in the T4P[26], leading to weakened glycan density upon applying the helical symmetry during cryo-EM map reconstructions.

Each pilin in the T4P has two disulfide bonds, involving cysteine/cysteine pairs (Cys57/Cys67 shown in Fig. 3d and Cys132/Cys145 shown in Fig. 3e). These disulfide bonds likely form when pilin subunits traverse through the oxidative periplasmic environment from the inner membrane pilin pool during pilus assembly. Surprisingly, a 13-Å gap in the density of the polypeptide chain is observed between Gly78 and Ser79 (Fig. 3b, c, f, and g). This indicates that, in addition to the post-translational cleavage at the N-terminus to remove the signal peptide involving the first seven amino acids in the prepilin, there is another cleavage in the middle of the peptide chain, a phenomenon not reported for other pilins. While the precise mechanism behind such pilin cleavage remains undiscovered, the fact that it takes place on the external surface of T4P suggests a direct influence on its interactions with the environment, such as its engagements with the immune system or phages.

### Structure of the AP205-T4P complex
To investigate the recognition mechanism between AP205 and T4P, we employed single-particle cryo-EM to obtain a structure of the AP205/T4P complex (Fig. 4a, b, and Supplementary Fig. 11). Both Mats within a virion can attach to a T4P individually through their apical domains,

with a higher likelihood of binding observed for the outer Mat (Supplementary Fig. 11a).

Within the T4P-Mat complex, the apical domain of each Mat specifically plugs into the groove formed by three adjacent pilin subunits within the T4P (referred to as Pilins 1, 2, and 3 in Fig. 4c). Notably, the C-terminal Ser147 of Pilins 1 and 2 are positioned near β3 and β7 of the Mat apical domain, respectively (Fig. 4d). Apparently, glycosylation at these two serines can prevent binding of AP205 Mat. It is likely that the substoichiometric glycosylation of pilins allows AP205 to bind to unglycosylated sites on the T4P. The smaller size of the apical domain further facilitates its fit into binding sites without potential hindrance from neighboring glycans on the surface of the pilus. We also observed that the internal cleavage site of Pilin 3 at Ser79 is located at the binding interface for the apical domain. This cleavage allows Ser79 to move ~13 Å away from Gly78, allowing its interaction with β8 of the Mat. The Mat of AP205 has evolved to adapt to such a post-translational cleavage at the phage-pilus binding interface and may not be compatible with an uncleaved pilin anymore (Supplementary Fig. 12).

### An AP205-derived protein scaffold targeting T4P
Our structures suggest that the Mat apical domain could be a protein scaffold that binds specifically to native *A. gp16* T4P. To create this scaffold, we cloned the N-terminal fragment of the Mat, encompassing the entire apical domain as well as β5, β6, and α2 from the central domain (Fig. 1c). This resulted in a continuous 200 amino acid sequence from the N-terminus, designated as Mat200. Mat200 was then recombinantly expressed in *E. coli* and purified to homogeneity (Supplementary Fig. 13). To assess the pilus binding ability of Mat200, we created a fusion of a super-fold green fluorescent protein to the C-terminus of Mat200 (Mat200-sfGFP). Using fluorescence microscopy,

we were able to visualize the binding of Mat$_{200}$-sfGFP to T4P in live *Acinetobacter* cells (Fig. 4e). This labeling process occurred rapidly, taking less than 5 minutes, and displayed highly specific binding. As a control, Coat$_{cb5}$-sfGFP, the coat protein from another ssRNA phage PhiCb5 fused to sfGFP, cannot bind to the same T4P.

## Discussion
Given that various bacterial T4P share a similar overall sequence and structure[28] (Supplementary Fig. 14), we may engineer the Mat$_{200}$ scaffold to bind to other T4P from other bacteria. Taking advantage of the fast mutation rate of ssRNA phages[29], one can use an evolution-and-selection approach to direct the Mat$_{200}$ scaffold to target other T4P. This could provide a promising diagnostic and therapeutic strategy for bacterial infections.

Merely 8% of AP205 virions exhibit a defined 3D conformation, representing a small subset among the total number of particles. The remaining 92% potentially assume more variable folds that pose challenges for cryo-EM image classifications or may stem from the inefficiencies in AP205 genome packaging. The presence of two Mat proteins within a single AP205 virion presents intriguing possibilities. An additional Mat could provide an expanded binding surface for the phage to attach to the receptor pili. Considering that *Acinetobacter* T4P exhibits substoichiometric glycosylation, the presence of an extra Mat protein could increase the chances of the phage locating an unglycosylated region on the pilus to bind. Furthermore, the gRNA is bound to each Mat via a conserved motif of "GCUUXGC" in Stem-loops 88 and 79 by forming similar π-stackings between RNA bases and aromatic sidechains of Mat. This suggests the gRNA has a similarly strong affinity for both Mats, with either of the two Mats capable of escorting the gRNA into the cell. Additionally, the binding of Mat-dimer to Stem-loops 88 and 79, which are approximately 500-nt apart in sequence, may facilitate the nucleation of the RNA, aiding the packaging process. Alternatively, the Mat-dimer offers an increased RNA binding sites, which may facilitate the recognition of the native AP205 gRNA during phage assembly.

The retraction of the receptor pili is believed to guide the Mat-bound gRNA through the channel within the pilus-assembly machine for cellular entry. In the natural environment, during the interaction of AP205 with *A.gp16*, it is plausible that multiple T4P from the same cell come into close proximity. This allows the pair of Mat proteins from a single phage to simultaneously bind to two pili (Supplementary Figs. 9c and 11), as shown in the major class of our single-particle cryo-EM structure of the in-vitro reconstituted AP205-pili complex. Clamping two neighboring pili to the same Mat-dimer might limit the independent extension/retraction of each pilus. The impact of this on pilus dynamics has yet to be characterized. Since each phage-bound pilus originates from its respective pilus-assembly machine (T4P basal body) at the cell envelope and will potentially compete for the bound phage particle. It remains to be determined under such circumstances, how the gRNA enters through the channel of the T4P basal body. Further experiments are necessary to address these intriguing questions.

## Methods
### Sample purifications and cryo-EM grid preparations
AP205 purification. 5 mL *A. gp16* overnight cell culture was incubated and shaken at 30 °C and 200 rpm. The 5 mL overnight culture was then inoculated into one 1 L culture, grown at 30 °C, and shaken at 200 rpm until the optical density (OD) ~ 0.1. 30 mL AP205 lysate (generated from previous purifications with a titer ~10$^6$) and 0.5 mM CaCl$_2$ were added to the 1 L culture, which was incubated for 30 min at room temperature. The 1 L culture was then shaken and incubated at 30 °C and 200 rpm overnight and spun down for 30 min at 3,470 g (4 °C). The cell pellet was resuspended in a buffer (20 mM Tris-HCl, 150 mM NaCl, pH 8.0). The resuspended cells were lysed

by passing through a microfluidizer at 25,000 psi. The total lysate was spun down at 41,657 g for 30 min at 4 °C and the supernatant was collected and concentrated to 1 mL. The 1 mL solution was added to a CsCl gradient solution (step gradient: 1.0, 1.2, 1.4, and 1.6 g/mL) and spun for 24 hr at 200,000 g using ultracentrifugation. The separated bands in the CsCl solution were drawn using syringes and dialyzed against a dialysis buffer (50 mM Tris-HCl, 150 mM NaCl, pH 8.0) to remove CsCl residues for 2 consecutive 24 hr at 4 °C. The fraction containing AP205 was confirmed by tittering against *A. gp16* cells, and the purity was checked by SDS-PAGE gel and negative-stain electron microscopy.

*A. gp16* T4P purification. 30 mL *A. gp16* overnight cell culture was incubated and shaken at 30 °C and 200 rpm. 100 μL of the overnight culture was added to each agar plate of the 100 plates made, spread using a cell spreader, and incubated at 30 °C overnight. The 100 overnight plates were then placed on ice and cells on plates were collected using a purification buffer (50 mM Tris-HCl, 150 mM NaCl, pH 8.0) and went through needles (diameter: 18 G + 25 G) 3 times to shear off the pili. The cell solution was spun down at 3470 g, 4 °C for 30 min to remove any remaining cells and the clear supernatant was collected and concentrated to 1 mL for a CsCl-gradient-based purification. The further purification steps were adopted from the AP205 purification protocol mentioned above, but the CsCl step gradient used was 1.0, 1.2, and 1.4 g/mL. The presence and purity of AP205 pili were confirmed by negative-stain electron microscopy.

Cryo-EM grid preparations. 3 μL of each purified sample, apo AP205, apo T4P, and AP205-T4P (by mixing AP205 and T4P at 1:1 volume ratio at 30 °C for 30 min), was applied to a 300-mesh 2/1 c-flat copper grid and vitrified with a Vitrobot Mark III (FEI Company) or a Vitrobot Mark IV (ThermoFisher) in vitreous ice.

### Single-particle cryo-EM
The numbers and parameters of data collections and processing for apo AP205, apo T4P, and AP205-T4P complex are summarized in Supplementary Table 1, respectively. The details are described below.

Data collection for the apo AP205. The samples of apo AP205 were imaged under a Titan Krios cryo-electron microscope (ThermoFisher) at SLAC National Accelerator Laboratory operated at 300 kV. 41,648 movie stacks were collected using EPU 1.2&2.1 with a Gatan K2 Summit direct detection camera (Gatan) at a nominal magnification of 130,000x in the counting mode, yielding a pixel size of 1.06 Å. A total electron dose of 50 e$^-$/Å$^2$ was fractionated over 35 frames (0.2 s/frame, 7 s in total). A post-column energy filter was applied with a slit width of 30 eV.

Data processing of the AP205 virion, *T=3*, and *T=4* VLPs. The movie stacks were first aligned and filtered by MotionCor2[30]. Particles with good quality were semi-automatically picked by EMAN2[31] or automatically picked using Gautomatch (http://www.mrc-lmb.cam.ac.uk/kzhang/). Gctf was used to estimate the contrast transfer function for each particle[32]. After 2D classification using RELION 3.0[33], 1,745,007 particles were selected for 3D Classification, which yielded 86,946 good particles with a defined gRNA conformation. To better resolve the gRNA, 65,745 particles from the dataset of the AP205-T4P complex (see Supplementary Fig. 1 and Supplementary Table 1 for more details) were also combined in the final refinement by cryoSPARC 3.0&4.0[34] to yield a reconstruction at a resolution of 3.1 Å. Only 152,691 (~8%) particles of AP205, corresponding to the mature virions, showed a genome with a defined tertiary structure.

Most of the particles showed an icosahedral capsid without a defined tertiary structure of the gRNA in the 3D classification result. Icosahedral refinement was then applied to these particles, yielding a *T=3* icosahedral structure of the capsid at 3.2-Å resolution, which presented features consistent with previously published structures of AP205 VLPs[16]. Around 1% of the particles had larger capsids (with a diameter of 310 Å). 7,716 of these larger particles were refined with

icosahedral symmetry and yielded a *T=4* icosahedral capsid at 3.4-Å resolution.

**Data collection of *A. gp16* T4P.** The samples of T4P were imaged under a Titan Krios G4 cryo-electron microscope (ThermoFisher) at the Texas A&M University operated at 300 kV. 5,946 movie stacks were collected using EPU with a Biocontinuum image-filtered Gatan K3 direct detection camera (Gatan) at a nominal magnification of 105,000x in the super-resolution mode, yielding a pixel size of 0.86 Å after binning by 2. A total electron dose of 50 e⁻/Å² was fractionated over 40 frames (0.2 s/frame, 8 s in total). A post-column energy filter was applied with a slit width of 15 eV.

**Data processing of *A. gp16* T4P.** Initial 2D references were generated from 1472 manually picked particles and these were used for the autopicking of 905,414 segments using CryoSPARC. From these segments, 2D class-averages with high-resolution details were identified, and the segments contributed to these averages were used for a helical reconstruction. Possible helical symmetries were estimated based on an averaged power spectrum generated from the raw particles and the outcome volumes were examined by trial and error. The data processing procedure is described in Supplementary Fig. 10.

**Data collection of the AP205-T4P complex.** Data were collected on the same cryo-electron microscope with the same set of parameters as apo AP205 except at different tilting angles to alleviate preferred orientations of the AP205-T4P complex on the grid[35]. In total, 12,703 movie stacks were obtained, with 3037, 1780, 7157, and 729 stacks collected at 0°, 15°, 30°, and 37° tilting, respectively.

**Data processing of the AP205-T4P complex.** The movie stacks were imported and processed using cryoSPARC. A total of 225,247 AP205 particles, regardless of T4P bound or not, were picked and subjected to multiple rounds of 2D classifications and heterogeneous refinements, yielding 94,095 particles classified as AP205-T4P complexes, which were subjected to one more round of 3D classification to generate three classes. Among these three classes, two classes showing two bound pili for each AP205 (~61 K particles) were selected and used for a homogeneous refinement of the AP205-Two-T4P complex at 3.6-Å resolution. Particles in the remaining class were subjected to an additional round of 3D classification into two classes. One of the two classes showing only one pilus bound to the outer Mat of AP205 ( ~ 17 K particles) was reconstructed at a resolution of 4.2 Å. The other class shows very weak pilus density and was discarded for further processing.

In these density maps of the AP205-T4P complexes, the T4P was poorly resolved, likely due to the flexibility of the pilus-bound apical domain of the Mat relative to the rest of the phage, leading to a spread of orientations for each T4P in the image particle. This caused inaccurate image alignments in cryo-EM data processing. To improve the density map, especially around the T4P region, we hierarchically classified the ~94 K particles. We first applied a generous mask around the regions of both outer and inner Mat, and the potentially associated T4P to locally classify all the particles into six classes. We then created another two tighter masks: one for the region covering only the outer Mat and its associated T4P while the other for the region covering only the inner Mat and its associated T4P. We locally refined the particles within each of the six classes from the previous round of 3D classification using each of the two tighter masks and obtained twelve reconstructions: six for the outer Mat-T4P complex and six for the inner Mat-T4P complex. The best reconstructions for the inner Mat-T4P and outer Mat-T4P complexes (out of 12 reconstructions) reached overall resolutions of 8.2 Å and 8.6 Å, respectively, but showed much more complete density for the T4P and clearer secondary structures of the pilin subunits. The density for the pilus that bound to the inner Mat had a better resolution compared to the one that bound to the outer Mat, as the apical domain of the inner Mat was stabilized by the basal domain of outer Mat (Helix α4, Fig. 1d) and the outer surface of the Coat shell (Supplementary Fig. 3. bottom left panel). Combining

particles in the six classes for either the inner or the outer Mat-T4P complexes did not improve the resolutions of the final refinements, which was probably due to the conformational heterogeneity in the long filamentous pili as we previously reported for the MS2-bound F-pili[15]. The data processing procedure is described in Supplementary Fig. 11.

### Molecular modeling and visualization

**AP205 mat modeling.** The initial model of Mat was generated by I-TASSER[36] and AlphaFold 2.2[37]. The inner and outer Mat models were then both manually fixed using Coot 0.8&0.9[38] and the ISOLDE[39] plugin in ChimeraX 1.5[40]. The final models were iteratively refined by RosettaCM[41] and Phenix 1.20.1[42] inside the cryo-EM density maps.

**AP205 gRNA modeling.** The RNA density was traced and segmented into fragments by UCSF Chimera 1.16[43]. The model for each fragment was generated and then connected using DRAFFTER[44]. The complete gRNA was then refined in the density using MDFF[45].

AP205 Coat shell modeling (including *T=3* and *T=4* VLPs). AP205 *T=3* VLP structure (PDB ID: 5LQP) was used to generate the Coat shell of AP205 mature virion. One Coat-dimer was removed to accommodate the Mat-dimer. The same strategy was used to generate the initial models for the capsids of AP205 *T=3* and *T=4* VLPs. These AP205 capsids, virions, and VLPs, were then fitted into the cryo-EM density maps using Chimera.

**AP205 virion modeling.** The models of AP205 Mat, Coat shell, and gRNA were combined in Chimera, iteratively refined using Phenix and manually inspected. Modifications were made using Coot and ISOLDE.

***A. gp16* T4P modeling.** *A. gp16* was whole-genome sequenced and the sequence of the type IV pilin was extracted (Supplementary Fig. 14). The first seven residues, corresponding to the signal peptide in the prepilin were removed. The remaining sequence was then submitted to AlphaFold to generate an initial model for the mature pilin, which is included in Supplementary Data 1 and was manually fitted into the density using Coot. The density of the first amino acid of the mature pilin (Phe8) is not complete in the cryo-EM map. Therefore, our model of the mature T4P pilin is from Thr9 to Ser147. In addition, the internal cleavage of the polypeptide was introduced between Gly78 and Ser79. This model was adjusted in ISOLDE and refined by Phenix iteratively. The model of the T4P filament was then generated by applying the helical symmetry on the pilin subunit.

**Mat-T4P complex modeling.** Within the local map of the T4P-bound inner Mat, we docked our high-resolution structures of the T4P and AP205 inner Mat from the virion, adjusted and refined the model of the complex using Coot, ISOLDE, and Phenix iteratively to model the details of the interaction between the Mat apical domain and the T4P (Supplementary Fig. 15). Such a model of the Mat-T4P complex is consistent with the locally refined density map of the T4P-bound outer Mat (Supplementary Fig. 15), even though at a lower resolution of 8.6 Å.

**AP205-T4P complex modeling.** To generate the AP205-T4P complex, the models of outer Mat-T4P and inner Mat-T4P complexes, previously refined in the local maps, were combined with the model of the apo AP205 virion by superimposing the corresponding Mat models.

### T4P-labeling by Mat₂₀₀-sfGFP

**Mat₂₀₀-sfGFP protein purification.** The His SUMO Mat₂₀₀-sfGFP sequence was codon optimized, synthesized, and inserted into a pET28a(+) vector by GenScript. Transformation of 100 ng pET28a(+) His SUMO Mat₂₀₀-sfGFP plasmid was performed on *E. Coli* BL21 DE3 cells, and 25 mL of overnight culture was added to 1 L of LB broth

supplemented with Kanamycin (final concentration: 50 μg/mL) for large-scale expressions. The 1 L cell culture was grown at 37 °C for approximately 4 hours and then cooled to 18 °C when the OD reached 0.6, followed by another 1-hour incubation. Protein expressions were induced using a final concentration of 0.5 mM IPTG and incubated for 16 hr. Cells were collected and centrifuged at 3470 g for 30 minutes. The cell pellet was resuspended in a lysis buffer (20 mM Tris-HCl, 150 mM NaCl, pH 8) containing an EDTA-free cocktail protease inhibitor tablet (ThermoFisher). The resuspended cells were lysed by passing through a microfluidizer three times at 25,000 psi. The total lysate was then centrifuged at 41,657 g for 30 minutes at 4 °C, and the supernatant was collected and injected into a Ni affinity column. The column was washed with 30 mL of washing buffer (20 mM Tris-HCl, 300 mM NaCl, pH 8, 5 mM Imidazole), and His SUMO Mat$_{200}$-sfGFP was eluted with 15 mL of elution buffer (20 mM Tris-HCl, 300 mM NaCl, pH 8, 300 mM Imidazole). The eluted protein was concentrated, and 500 μL of the concentrated protein was loaded onto a pre-equilibrated Superdex 200 size-exclusion column using an equilibration buffer (20 mM Tris-HCl, 300 mM NaCl, pH 8). The fraction which corresponds to the SUMO Mat$_{200}$-sfGFP protein was collected.

To remove the SUMO tag, SUMO protease was added at a concentration of 1 mg/mL to SUMO Mat$_{200}$-sfGFP, and the volume was adjusted to 50 mL. The SUMO protease cleavage reaction was carried out at room temperature for 1 hr and confirmed success by SDS-PAGE gel analysis. The monomeric form of Mat$_{200}$-sfGFP was further purified by the size exclusion column (Supplementary Fig. 13).

Fluorescence microscopy analysis. Mat$_{200}$-sfGFP was adjusted to a working concentration, 1.5 μM. The over-day culture of *A. gp16* cells were grown to mid-exponential phase before being aliquoted and mixed with diluted Mat$_{200}$-sfGFP at a 1:1 ratio. Samples were incubated at room temperature for 5 minutes, and a small drop (~1.5 μL) of the mixture was added onto a PBS-agarose pad before applying a coverslip on top for imaging. Images were taken as Z-series with 300 nm interval in both GFP and phase contrast channels.

For the cells used in the study, the information is presented in Supplementary Table 2.

## Reporting summary

Further information on research design is available in the Nature Portfolio Reporting Summary linked to this article.

## Data availability

The coordinates of *A. gp16* T4P, AP205, inner Mat-T4P complex, outer Mat-T4P complex, AP205 *T=4* VLP, and AP205 *T=3* VLP are deposited in the Protein Data Bank (PDB) with the accession codes 8TOB, 8TOC, 8TV9, 8TVA, 8TW2, and 8TWC, respectively, as well as the referenced AP205 *T=3* VLP has the PDB code 5LQP. Cryo-EM maps of *A. gp16* T4P alone, AP205 alone, inner Mat-T4P complex, outer Mat-T4P complex, AP205 *T=4* VLP, AP205 *T=3* VLP, AP205 bound to two T4P, and AP205 bound to one T4P are deposited in the Electron Microscopy Data Bank (EMDB) with the accession codes for EMD-41442, EMD-41443, EMD-41634, EMD-41635, EMD-41657, EMD-41666, EMD-41646, and EMD-41447, respectively. Source data are provided with this paper.

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

## Acknowledgements
We acknowledge the Microscopy and Imaging Center at Texas A&M University for providing instrumentation for the initial screening of the cryo-EM samples, the Texas A&M High-Performance Research Computing Center for computational resources for data processing, Dr. Gaya Yadav at the BCBP cryo-EM center at Texas A&M University, and the cryo-EM regional data collection consortium at SLAC for high-resolution single-particle cryo-EM data collection. We also thank for these supporting grants, Center for Phage Technology fund, TAMU (L.Z., J.Z.), National Institutes of Health grant R01GM141659 (L.Z., J.Z.), National Institutes of Health grant R21AI156846 (L.Z., J.Z.), National Institutes of Health grant U24GM1167 (J.Z.), National Science Foundation grant MCB-1902392 (L.Z., J.Z.), and TAMU T3 and X-grants (L.Z., J.Z.).

## Author contributions
R.M., Z.X. J.C., and J.Z. contributed to the conceptualization. J.C. and F.W. contributed to the methodology. R.M., Z.X., J.C., Z.Y., J.T., W.X., Y.W., K.C., Z.Z., F.W., R.Y., L.Z. and J.Z. contributed to the investigation. R.M., Z.X., J.C., J.T., Z.Y., and J.Z. contributed to the visualization. L.Z. and J.Z. contributed to the funding acquisition. J.Z. contributed to the project administration. R.Y., L.Z. and J.Z. contributed to the supervision. All authors contributed to the manuscript writing. These authors contributed equally: R.M., Z.X. and J.C.

## Competing interests
Z.X., Z.Y., L.Z. and J.Z. have filed a provisional patent on Mat$_{200}$ through Texas A&M University (Application No. 63/583763). J.Z. serves as a consultant for Lynntech Inc. College Station, TX. All other authors declare no competing interests.
