## [Peer Review File · Nature Communications]

Structural basis of *Acinetobacter* type IV pili targeting by an RNA virusREVIEWER COMMENTS

Reviewer #1 (Remarks to the Author):

This manuscript by Meng et al. describes the structure of the virion of an Acinetobacter-targeting RNA-virus alone and a high-resolution reconstruction of the type IV pili from its Acinetobacter host, individually and in complex. The elucidation of this complex represents a substantive advance in our understanding of the mechanisms of this host/phage interaction as well as providing new insights into other aspects of both the virus and host organism through the virion and pilus structures. The structure of the AP205 virion and the Acinetobacter gp16 type IV pilus are both resolved to high resolution, providing useful insights into their biogenesis (ex. the proteolytic cleavage of the PilA subunits) and the complex of the AP205 virion with the T4P provides useful advances in understanding the structural mechanism of phage infection (that the major pilin subunit is the receptor and that two pili can be bound simultaneously). My specific comments (below) are mostly related to the details of that interface, where it appears the data quality may somewhat limit our ability to draw conclusions; making the most of that data and communicating to the reader where ambiguity exists will put the finishing touches on what is, in my mind, an excellent manuscript.

Specific Comments:

- One of the key advances reported here is the structure of the Mat-T4P complex, but as the authors describe it, the molecular description of the interface is limited by the quality of the reconstruction. To appreciate this, the readers really need a view of the density in this region; Figure 4c,d do not provide sufficient detail for the reader to draw any conclusions about, as the title of the paper attests, the structural basis for this interaction. More detail is needed and (perhaps as an extended figure) density needs to be included to give the reader some indication of the degree of certainty.
- Do the authors have any data indicating what percentage of PilA subunits are glycosylated? I agree with the authors that sporadic/incomplete glycosylation is a good explanation for the poor density of the glycan, but other explanations (including conformational or even constitutional heterogeneity) are equally possible. The authors describe Extended Figure 6 as showing binding to glycosylated pili, but can we be certain that the binding is to glycosylated subunits? In lines 234-235, the authors appear to claim the opposite. I think this is an important point in terms of the mechanism of phage infection and the evolutionary relationship between c-terminal glycosylation of the pilin and phage infection and Figure 4D, showing the model without density does not really give the reader a clear idea of whether the data is clear enough to rule out glycosylation of these pilins.
- Related to the above, the statement "Previous studies have indicated that glycosylation of pili assists Acinetobacter cells in evading host immunity and certain dsDNA bacteriophages" would be a well-supported statement for glycosylation in *Pseudomonas aeruginosa* T4P, but I am less certain the data is there for Acinetobacter. Reference 26 shows that the non-glycosylated T4P mutant forms less biofilm and shows reduced virulence in two model host organisms. No difference is reported related to phage-

resistance or (directly) to host immunity. Data on natural competence and host adherence for an S136A non-glycosylated mutant of PilA from *A. nosocomialis* M2 (formerly *A. baumannii* M2) is available in reference 25 and here: <https://onlinelibrary.wiley.com/doi/10.1111/mmi.12986>

- Are the inner Mat-T4P and outer Mat-T4P complexes distinct in terms of the docking angle or some other criterion? If I am reading lines 492-497 correctly, the differential classification of inner Mat-T4P and outer Mat-T4P complexes was done manually; if superimposed, are there meaningful differences between the two?

- Some minor potential grammatical issues (Lines 19-20, I'm not sure "Acinetobacters" is correct, rather than 'Acinetobacter bacteria' or 'Acinetobacter species', Line 25, 706; I don't know that "adsorbed" is used correctly here)

- For Extended Figure 8, I may be misunderstanding the way the resolution estimations are depicted in the figure, but if the locally-refined T4P-Mat complexes are at a resolution of ~8Å (by GSFSC), how can the corresponding region in the globally-refined particle reconstruction be marked at ~5.5Å?

Reviewer #2 (Remarks to the Author):

In this study, the authors present their structure of *Acinetobacter* ssRNA bacteriophage AP205, the structure of type IV pili from the host *Acinetobacter*, alone, and the complex of the phage bound to the pilus. In addition, they show binding of a recombinant maturation protein fragment to *Acinetobacter* cells. The structures show some interesting differences from the classical ssRNA phage systems that have been studied previously, MS2 and Qbeta. The structures are of high quality, the study is generally well presented and is nicely illustrated. I have no major concerns about the paper, but a few minor issues are listed below:

Line 19, 37, 40, 63: When used as a general term for the genus in the plural, "acinetobacters" should not be capitalized or italicized.

Line 21: "target bacteria via retractive pili" is unclear. Better to write "target the retractive pili of host bacteria" or something like that.

On lines 64-70, it difficult to understand what is prior knowledge and what was gained in the present study. "currently, the only..." should rather be stated as "The structure of AP205 coat was solved previously..." Instead of "The complete structure ... remains unknown" write "Here, we have determined the complete structure of ..." or something similar.

Line 75: "Only 8% of the particles showed a defined conformation." I guess it could be more accurately stated that only 8% showed a consistent conformation. It is quite possible that the RNA was ordered in the remaining 92%, but it different ways. Do we know whether the 8% with a consistent conformation are more biologically relevant than the other particles? The authors could probably be a bit more circumspect in that in reality they are solving a relatively small subset of the possible virion structures.

Furthermore, did the icosahedral reconstructions yield any additional information? I'm puzzled why the T=3 icosahedral reconstruction did not reach as high resolution as the asymmetric reconstruction; one would expect it to be better.

Could the author explain what they mean by "circular permutation" of the coat protein structure? (I realize this was previously published, but some explanation would be helpful.) Did the new icosahedral reconstruction reveal any new details not in the previous publication?

Line 93 and 180: Delete "other ssRNA phages, like/such as," since this statement is probably too general. Are there other ssRNA phages that these statements also apply to?

Line 185: should there be a species epithet after "Acinetobacter"? Perhaps this is an unspecified species? Please provide a database reference to the sequence.

The pilus-binding activity of the recombinant protein is interesting. It would be even more interesting to know whether it causes pilus retraction and/or affect cell viability or virulence.

The discussion is very short. Conciseness is a virtue, but it might be nice to expand it somewhat to include a broader discussion of the AP205 structure and the T4 pilus, their relationship to other structure, the process of assembly, genome incorporation and infection.

Reviewer #3 (Remarks to the Author):

The manuscript in my opinion is outstanding and unveils significant findings on the cryo-EM structure of the ssRNA AP205 virion, the T4P structure of *Acinetobacter* species genotype 16 (A. gp16), and the complex formed by the ssRNA AP205 virion with the T4P filament. Additionally, it details a 20-kilodalton protein derived from the AP205 virion that targets the A. gp16 T4P filament in situ.

The substantial results contribute significantly to the field, marking a noteworthy achievement as it is, to the best of my knowledge, the first instance of a T4P filament featuring post-translation pilin cleavage between G78-S79 and the MAT protein of the phage demonstrates adaptation to this modification. Moreover, building upon the insights gained from the interaction between this phage and the TP4 pilus, the research group engineered a protein derived from the phage. This protein exhibits the capability to interact with the filament, demonstrating promising potential for identifying the A. gp16 filament.

Here are some questions that were not clear when reading the manuscript:

1- In the abstract, it is mentioned that the 20kDa protein has potential for diagnosis. Do you have evidence that this phage would recognize other strains of *A. baumannii*? Or is it specific to A. gp16?

2- Is the observed cleavage between residues G78-S79 expected in other type IV pili? An analysis of amino acid conservation could provide insight into whether this cleavage is an exception or more widely found in nature.

3- Has the G78-S79 cleavage been demonstrated by another biophysical technique or only visualized in the Cryo-EM map?

4- In the second paragraph of the introduction, could you briefly discuss the evolutionary relationship of T4P with other systems like T2SS, among others?

5- It is not clear whether the phage is lysogenic or lytic; I think it would be worth mentioning in the introduction.

6- In line 81, it is written 42669-nt gRNA. The presentation seems confusing, but it appears to be the size of the phage's genomic RNA?

7- In line 89, it is described that there is a region in the viral gRNA that is exposed and interacts with the MAT protein dimer, and that this interaction should be due to the GCUUXGC sequence. Would it be possible to prove this hypothesis by measuring the interaction of MAT with this sequence, or does the interaction only occur in the context of the viral particle?

8- In line 250, Supplementary Fig. 4 is mentioned, but I could not find this figure in the manuscript.

9- In lines 261-262, it is suggested that the Mat200 protein would bind to other T4Ps. Do you have experimental evidence that this would occur? As far as I know, phages are generally specific to each bacterial species.

10- The sentence located in lines 284-286 is a bit confusing. It is not clear to me what is meant by "channel of the pilus bound to the outer Mat."

11- In lines 565-566, how was it demonstrated that the protein is monomeric? Was SEC-MALS performed, or was the SEC column calibrated and the protein MW calculated subsequently?

12- In line 571, Supplementary Fig.4 is mentioned, but I could not find it in the manuscript.

13- In line 659, I am curious to know whether the shown structure assembles on its own or requires accessory proteins.

REVIEWER COMMENTS

Reviewer #1 (Remarks to the Author):

This manuscript by Meng et al. describes the structure of the virion of an Acinetobacter-targeting RNA-virus alone and a high-resolution reconstruction of the type IV pili from its Acinetobacter host, individually and in complex. The elucidation of this complex represents a substantive advance in our understanding of the mechanisms of this host/phage interaction as well as providing new insights into other aspects of both the virus and host organism through the virion and pilus structures. The structure of the AP205 virion and the Acinetobacter gp16 type IV pilus are both resolved to high resolution, providing useful insights into their biogenesis (ex. the proteolytic cleavage of the PilA subunits) and the complex of the AP205 virion with the T4P provides useful advances in understanding the structural mechanism of phage infection (that the major pilin subunit is the receptor and that two pili can be bound simultaneously). My specific comments (below) are mostly related to the details of that interface, where it appears the data quality may somewhat limit our ability to draw conclusions; making the most of that data and communicating to the reader where ambiguity exists will put the finishing touches on what is, in my mind, an excellent manuscript.

Reviewer #1 comments

Reviewer's comment	One of the key advances reported here is the structure of the Mat-T4P complex, but as the authors describe it, the molecular description of the interface is limited by the quality of the reconstruction. To appreciate this, the readers really need a view of the density in this region; Figure 4c,d do not provide sufficient detail for the reader to draw any conclusions about, as the title of the paper attests, the structural basis for this interaction. More detail is needed and (perhaps as an extended figure) density needs to be included to give the reader some indication of the degree of certainty.
Author's response	We thank the reviewer for the comment. We have revised the Supplementary Fig. 15 (see the figure below) to show both the model and density at the Mat-pilus interfaces.  Supplementary Fig. 15. a The atomic models of inner and outer Mat-pilus

	complexes are shown fitted into their respective locally refined cryo-EM density maps. Black arrows indicate the Mat-pilus interfaces. Although the two locally refined maps are at resolutions $\sim 8\text{\AA}$, we can still clearly see the protein secondary structures, such as the protein helices, matching the sausage-like densities (Panel a). We took extra care to model and refine the Mat-pilus interfaces using Rosetta, ISOLDE, and Phenix iteratively. Such a strategy was applied to both the outer Mat-T4P complex and the inner Mat-T4P complex independently, allowing us to cross-validate our models.
Reviewer's comment	Do the authors have any data indicating what percentage of PilA subunits are glycosylated? I agree with the authors that sporadic/incomplete glycosylation is a good explanation for the poor density of the glycan, but other explanations (including conformational or even constitutional heterogeneity) are equally possible. The authors describe Extended Figure 6 as showing binding to glycosylated pili, but can we be certain that the binding is to glycosylated subunits? In lines 234-235, the authors appear to claim the opposite. I think this is an important point in terms of the mechanism of phage infection and the evolutionary relationship between c-terminal glycosylation of the pilin and phage infection and Figure 4D, showing the model without density does not really give the reader a clear idea of whether the data is clear enough to rule out glycosylation of these pilins.
Author's response	We appreciate the comment from the reviewer. The weaker density for the glycans can be attributed to the sporadic glycosylation as well as the conformational and constitutional heterogeneity in the glycans attached to Ser147. We have revised the text in the manuscript accordingly. “The cryo-EM density of the glycan is weaker than the protein density. This might be attributed to the flexible attachment of the glycan to Ser147, glycan constitutional heterogeneity, and substoichiometric glycosylation of pilins in the T4P, leading to weakened glycan density upon applying the helical symmetry during cryo-EM map reconstructions.” Currently, we don't know the percentage of PilA subunits that are glycosylated. This is an ongoing experiment using mass-spec that we plan to publish in the future. In the original Extended Figure 6 (now Supplementary Fig. 9), we show that AP205 can bind to T4P, presumably with some but not all pilins being glycosylated. Therefore, AP205 can bind around the pilin subunits which are not glycosylated. This is consistent with the observation in Supplementary Fig. 9 Panel b that phage particles are sparsely decorating the straight T4P filaments, presumably to the unglycosylated pilins. To more clearly illustrate the fact that AP205 cannot bind to the T4P binding sites that have the neighboring PilA subunits glycosylated, we have made a new

	Panel b in Supplementary Fig. 15, which shows the density of the glycans will obstruct the binding of the Mat apical domain (see figure below).  Supplementary Fig. 15. b The model of Inner Mat-pilus complex is fitted into the reconstructed A. gp16 T4P Cryo-EM density map. Glycan densities (gray) from the Cryo-EM map are highlighted in red boxes. It is observed that the presence of glycans on the T4P will interfere with the binding of the Mat (purple model).
--	---

Reviewer's comment	Related to the above, the statement “Previous studies have indicated that glycosylation of pili assists Acinetobacter cells in evading host immunity and certain dsDNA bacteriophages” would be a well-supported statement for glycosylation in Pseudomonas aeruginosa T4P, but I am less certain the data is there for Acinetobacter. Reference 26 shows that the non-glycosylated T4P mutant forms less biofilm and shows reduced virulence in two model host organisms. No difference is reported related to phage-resistance or (directly) to host immunity. Data on natural competence and host adherence for an S136A non-glycosylated mutant of PilA from A. nosocomialis M2 (formerly A. baumannii M2) is available in reference 25 and here: https://onlinelibrary.wiley.com/doi/10.1111/mmi.12986
Author's response	We are thankful for the reviewer's comment. We have revised the text by removing the mentioning of phage-resistance or host immunity to a more appropriate statement below. “These glycans are added post-translationally to the last amino acid Ser147 of the mature pilin. Previous studies have indicated that glycosylation of pili assists A. baumannii in biofilm formation and virulence.”

Reviewer's comment	Are the inner Mat-T4P and outer Mat-T4P complexes distinct in terms of the docking angle or some other criterion? If I am reading lines 492-497 correctly, the differential classification of inner Mat-T4P and outer Mat-T4P complexes was done manually; if superimposed, are there meaningful differences between the two?
Author's response	We thank the reviewer for the comments. When we superimpose the models for

	the T4P filaments in the outer Mat-T4P complex and the inner Mat-T4P complex, the RMSD of the apical domains of the outer and the inner Mat proteins are less than $\sim 1.5 \text{ \AA}$, which is indistinguishable at the current resolutions of our locally refined maps. Please see the figure below.  Panel a. Models of the inner Mat-T4P and the outer Mat-T4P complex with the respective T4P aligned to show the differences in the inner and out Mat proteins. Panel b and c. Zoom-in views of the Mat and Mat apical domains in Panel a.
--	--

Reviewer's comment	Some minor potential grammatical issues (Lines 19-20, I'm not sure "Acinetobacters" is correct, rather than 'Acinetobacter bacteria' or 'Acinetobacter species', Line 25, 706; I don't know that "adsorbed" is used correctly here)
Author's response	We thank the review for the comment. As the other reviewer pointed out, we have changed it to the plural form without being capitalized or italicized. And we also reworded the 'adsorbed' to 'bound'.

Reviewer's comment	For Extended Figure 8, I may be misunderstanding the way the resolution estimations are depicted in the figure, but if the locally-refined T4P-Mat complexes are at a resolution of $\sim 8 \text{ \AA}$ (by GSFSC), how can the corresponding region in the globally-refined particle reconstruction be marked at $\sim 5.5 \text{ \AA}$?
Author's response	We thank the reviewer for the comments. We have revised the original Extended Figure 8 (now Supplementary Figure 11) to avoid any confusion. The range of the resolution scale bar is updated to reflect a broader resolution range.

Reviewer #2 (Remarks to the Author):

In this study, the authors present their structure of Acinetobacter ssRNA bacteriophage AP205, the structure of type IV pili from the host Acinetobacter, alone, and the complex of the phage bound to the pilus. In addition, they show binding of a recombinant maturation protein fragment to Acinetobacter cells. The structures show some interesting differences from the classical ssRNA phage systems that have been studied previously, MS2 and Qbeta. The structures are of high quality, the study is generally well presented and is nicely illustrated. I have no major concerns about the paper, but a few minor issues are listed below:

Reviewer #2 comments

Reviewer's comment	Line 19, 37, 40, 63: When used as a general term for the genus in the plural, "acinetobacters" should not be capitalized or italicized.
Author's response	We appreciate the reviewer's comment. We have corrected this in the manuscript.

Reviewer's comment	Line 21: "target bacteria via retractive pili" is unclear. Better to write "target the retractive pili of host bacteria" or something like that.
Author's response	We thank the reviewer for the comment. We have reworded the manuscript as the reviewer suggested to "target the retractile bacterial pili".

Reviewer's comment	On lines 64-70, it difficult to understand what is prior knowledge and what was gained in the present study. "currently, the only..." should rather be stated as "The structure of AP205 coat was solved previously..." Instead of "The complete structure ... remains unknown" write "Here, we have determined the complete structure of ..." or something similar.
Author's response	We thank the reviewer for the comment. We have rephrased the manuscript as the reviewer suggested.

Reviewer's comment	Line 75: "Only 8% of the particles showed a defined conformation." I guess it could be more accurately stated that only 8% showed a consistent conformation. It is quite possible that the RNA was ordered in the remaining 92%, but it different ways. Do we know whether the 8% with a consistent conformation are more biologically relevant than the other particles? The authors could probably be a bit more circumspect in that in reality they are solving a relatively small subset of the possible virion structures.
Author's response	We agree with the reviewer. To be more circumspect, we add this in the paper's discussion. Currently we do have an ongoing project in the lab to determine

	whether a defined conformation is more biologically relevant, e.g. more infectious, and hopefully can get an answer in the near future. The following text is added. “Merely 8% of AP205 virions exhibit a defined 3D conformation, representing a small subset among the total number of particles. The remaining 92% potentially assume more variable folds that pose challenges for cryo-EM image classifications or may stem from the inefficiencies in AP205 genome packaging.”

Reviewer’s comment	Furthermore, did the icosahedral reconstructions yield any additional information? I’m puzzled why the T=3 icosahedral reconstruction did not reach as high resolution as the asymmetric reconstruction; one would expect it to be better.
Author’s response	We appreciate the reviewer's comment. For the $T=3$ icosahedral reconstructions, it is consistent with the previously published structure, albeit at a higher resolution. The overall resolutions for the $T=3$ and asymmetric reconstructions are estimated by FSC at 3.2Å and 3.1Å, respectively. Intuitively, one would expect a $T=3$ reconstruction would produce higher Fourier shell correlation as more asymmetric units are averaged in the $T=3$ reconstruction. One explanation might be that the $T=3$ capsid still contains structural features or heterogeneity that cannot be described by a perfect icosahedral symmetry, such as the asymmetrically bound and heterogeneous RNA in these VLPs. Additionally, the asymmetric reconstruction incorporated an additional dataset which may have even better quality, giving rise to a slightly better reported FSC.

Reviewer’s comment	Could the author explain what they mean by “circular permutation” of the coat protein structure? (I realize this was previously published, but some explanation would be helpful.) Did the new icosahedral reconstruction reveal any new details not in the previous publication?
Author’s response	A circular permutation is a relationship between proteins whereby the proteins have a changed order of amino acids in their peptide sequence. The result is a protein structure with different connectivity, but overall similar three-dimensional (3D) shape. We have revised the text to explain circular permutation as shown below. “Currently, only the structure of AP205 Coat has been solved, revealing a circular permutation compared to the Coat of MS2 and Qβ, whereby the proteins have a changed order of amino acids in their peptide sequences, while the overall three-dimensional shapes are similar.”

Reviewer's comment	Line 93 and 180: Delete “other ssRNA phages, like/such as,” since this statement is probably too general. Are there other ssRNA phages that these statements also apply to?
Author's response	We thank the reviewer for the comment. We took the reviewer's suggestion and deleted the sentences. As for other ssRNA phages, we have not found any.

Reviewer's comment	Line 185: should there be a species epithet after “Acinetobacter”? Perhaps this is an unspecified species? Please provide a database reference to the sequence.
Author's response	We thank the reviewer for the comment. The host should be Acinetobacter genomosp. 16 (we abbreviate it as A. gp16). And here is the NCBI data bank reference link. NCBI:txid70347 https://www.ncbi.nlm.nih.gov/Taxonomy/Browser/wwwtax.cgi?mode=Info&id=70347

Reviewer's comment	The pilus-binding activity of the recombinant protein is interesting. It would be even more interesting to know whether it causes pilus retraction and/or affect cell viability or virulence.
Author's response	We agree with the reviewer that Mat ₂₀₀ is an interesting protein that can potentially change the pilus retraction dynamics and affect the host virulence. This further investigation is currently on-going in the lab for a future publication.

Reviewer's comment	The discussion is very short. Conciseness is a virtue, but it might be nice to expand it somewhat to include a broader discussion of the AP205 structure and the T4 pilus, their relationship to other structures, the process of assembly, genome incorporation and infection.
Author's response	We thank the review for the comments and have expanded our discussion.

Reviewer #3 (Remarks to the Author):

The manuscript in my opinion is outstanding and unveils significant findings on the cryo-EM structure of the ssRNA AP205 virion, the T4P structure of *Acinetobacter* species genotype 16 (*A. gp16*), and the complex formed by the ssRNA AP205 virion with the T4P filament. Additionally, it details a 20-kilodalton protein derived from the AP205 virion that targets the *A. gp16* T4P filament in situ. The substantial results contribute significantly to the field, marking a noteworthy achievement as it is, to the best of my knowledge, the first instance of a T4P filament featuring post-translational pilin cleavage between G78-S79 and the MAT protein of the phage demonstrates adaptation to this modification.

Moreover, building upon the insights gained from the interaction between this phage and the TP4 pilus, the research group engineered a protein derived from the phage. This protein exhibits the capability to interact with the filament, demonstrating promising potential for identifying the *A. gp16* filament.

Reviewer #3 comments

Reviewer's comment	1- In the abstract, it is mentioned that the 20kDa protein has potential for diagnosis. Do you have evidence that this phage would recognize other strains of A. baumannii ? Or is it specific to A. gp16 ?
Author's response	We thank the reviewer for the comment. AP205 is specific to A. gp16. As an on-going project in the lab, we have displayed the Mat₂₀₀ of AP205 on the M13 phage display system, which would allow us to evolve and select Mat₂₀₀ variants to bind to other acinetobacter T4P. Given the high homology between different T4P, we would expect it is feasible to find a Mat₂₀₀ variant that would bind to T4P of other strains.

Reviewer's comment	2- Is the observed cleavage between residues G78-S79 expected in other type IV pili? An analysis of amino acid conservation could provide insight into whether this cleavage is an exception or more widely found in nature.
Author's response	The cleavage between residue G78-S79 is not expected in other T4P. As shown by Supplementary Fig. 14, we aligned Acinetobacter baylyi, Acinetobacter baumannii, and Pseudomonas pilin sequences (Panel a) and compared the pilin structures (panel b). It seems that the other T4P pilins lack such a cleavage motif.

Reviewer's comment	3- Has the G78-S79 cleavage been demonstrated by another biophysical technique or only visualized in the Cryo-EM map?
Author's response	The cleavage has only been visualized by cryo-EM of native T4P purified from the cells.

Reviewer's comment	4- In the second paragraph of the introduction, could you briefly discuss the evolutionary relationship of T4P with other systems like T2SS, among others?
--

Author's response	We appreciate the reviewer for the comment and have revised the introduction as follows. “T4P is assembled by the type IV pilus system (T4PS), which is homologous to the type II secretion system (T2SS). While T4PS is responsible for assembling T4P, T2SS assembles a pseudopilus assumed to function as a 'piston' to facilitate protein secretion.”
-------------------	---

Reviewer's comment	5- It is not clear whether the phage is lysogenic or lytic; I think it would be worth mentioning in the introduction.
---

Author's response	AP205 is an ssRNA phage, which is different from DNA phages that have lytic and lysogenic cycles. ssRNA phages are lytic phages. We have added this in the introduction. “They are lytic phages and their genomes of these ssRNA viruses, typically 3-4kb long, exhibit a conserved organization of three core genes.”
-------------------	--

Reviewer's comment	6- In line 81, it is written 42669-nt gRNA. The presentation seems confusing, but it appears to be the size of the phage's genomic RNA?
---

Author's response	We thank the reviewer for the comment. Yes, the gRNA consists of 4,269 nucleotides. We will rephrase the sentence to avoid any confusion.
-------------------	---

Reviewer's comment	7- In line 89, it is described that there is a region in the viral gRNA that is exposed and interacts with the MAT protein dimer, and that this interaction should be due to the GCUUXGC sequence. Would it be possible to prove this hypothesis by measuring the interaction of MAT with this sequence, or does the interaction only occur in the context of the viral particle?
---

Author's response	We thank the reviewer for the comment. We have attempted to purify the maturation protein recombinantly, but the protein solubilities seem to be an issue, which is a common problem for ssRNA maturation proteins. Consequently, quantitative analysis in vitro for the binding of purified Mat and this gRNA is still an on-going work.
-------------------	---

Reviewer's comment	8- In line 250, Supplementary Fig. 4 is mentioned, but I could not find this figure in the manuscript.
Author's response	We thank the reviewer for the comment. For the first submission, all supplementary figures are in a separate PDF file. For this revised submission, we will combine the supplements with the main manuscript and extended data figures into one single pdf. The original Supplementary Fig. 4 (now Supplementary Fig. 13) is shown below. 
Reviewer's comment	9- In lines 261-262, it is suggested that the Mat200 protein would bind to other T4Ps. Do you have experimental evidence that this would occur? As far as I know, phages are generally specific to each bacterial species.
Author's response	We thank the reviewer for the comment. And we agree that phages are specific to their bacterial species. In lines 261-262, we were proposing that protein evolution can be performed on the Mat₂₀₀ protein to make the protein selectively target other bacterial T4Ps. We have encoded Mat₂₀₀ in a M13 phage display system. We are now evolving Mat₂₀₀ to target other Acinetobacter species, which will be presented in the future.

Reviewer's comment	10- The sentence located in lines 284-286 is a bit confusing. It is not clear to me what is meant by "channel of the pilus bound to the outer Mat."
Author's response	We appreciate the reviewer's comment. We reworded and changed the discussion in the main manuscript as shown below. Regarding the "channel of

	the pilus bound to the outer Mat”, it is referring the channel within the pilus-assembly machine, which is part the T4P basal body. “Clamping two neighboring pili to the same Mat-dimer might limit the independent extension/retraction of each pilus. The impact of this on pilus dynamics has yet to be characterized. Since each phage-bound pilus originates from its respective pilus-assembly machine (T4P basal body) at the cell envelope and will potentially compete for the bound phage particle. It remains to be determined under such circumstances, how the gRNA enters through the channel of the T4P basal body. Further experiments are necessary to address these intriguing questions.”
--	---

Reviewer’s comment	11- In lines 565-566, how was it demonstrated that the protein is monomeric? Was SEC-MALS performed, or was the SEC column calibrated and the protein MW calculated subsequently?
Author’s response	Yes. The SEC column is calibrated, and protein MW is calculated subsequently. After removing the SUMO tag, we run the SEC column again to further characterize the molecular weight to be consistent with a monomer.

Reviewer’s comment	12- In line 571, Supplementary Fig.4 is mentioned, but I could not find it in the manuscript.
Author’s response	We thank the reviewer for the comment. For the first submission, we do have a separate pdf for all supplementary figures, which included supplementary figure 4. For this new submission, we combined these files into one single pdf. The original Supplementary Fig. 4 is now Supplementary Fig. 13. Please see below.

Reviewer's comment	13- In line 659, I am curious to know whether the shown structure assembles on its own or requires accessory proteins.
Author's response	The genomic RNA (gRNA) folds on its own thermodynamically. However, due to the large size of the entire gRNA, assembling it into a well-defined structure at minimal energy remains difficult. Therefore, the co-assembly of the capsid proteins around the gRNA is required to guide the gRNA to fold correctly.

REVIEWERS' COMMENTS

Reviewer #1 (Remarks to the Author):

These additions address my concerns.

Reviewer #2 (Remarks to the Author):

In this revised manuscript, the authors have addressed the minor concerns of all three reviewers, including myself. I have no further concerns or comments.

Reviewer #3 (Remarks to the Author):

All questions have being addressed, and I am content with the responses provided. I think the manuscript is ready to be published.